# Development of 16 novel EST-SSR markers for species identification and cross-genus amplification in sambar, sika, and red deer

Chen Hsiao[1], Hsin-Hung Lin[2], Shann-Ren Kang[2], Chien-Yi Hung[1], Pei-Yu Sun[1], Chieh-Cheng Yu[1], Kok-Lin Toh[1], Pei-Ju Yu[1], Yu-Ten Ju[1]*

**1** Department of Animal Science and Technology, National Taiwan University, Taipei, Taiwan, **2** Kaohsiung Animal Propagation Station, Pingdong, Taiwan

* ytju@ntu.edu.tw

**Data Availability Statement:** All relevant data are within the manuscript and its Supporting Information files.

## Abstract

Deer genera around the globe are threatened by anthropogenic interference. The translocation of alien species and their subsequent genetic introgression into indigenous deer populations is particularly harmful to the species of greatest conservation concern. Products derived from deer, including venison and antler velvet, are also at risk of fraudulent labeling. The current molecular markers used to genetically identify deer species were developed from genome sequences and have limited applicability for cross-species amplification. The absence of efficacious diagnostic techniques for identifying deer species has hampered conservation and wildlife crime investigation efforts. Expressed sequence tag-simple sequence repeat (EST-SSR) markers are reliable tools for individual and species identification, especially in terms of cross-species genotyping. We conducted transcriptome sequencing of sambar (*Rusa unicolor*) antler velvet and acquired 11,190 EST-SSRs from 65,074 newly assembled unigenes. We identified a total of 55 unambiguous amplicons in sambar (n = 45), which were selected as markers to evaluate cross-species genotyping in sika deer (*Cervus nippon*, n = 30) and red deer (*Cervus elaphus*, n = 46), resulting in cross-species amplification rates of 94.5% and 89.1%, respectively. Based on polymorphic information content (>0.25) and genotyping fidelity, we selected 16 of these EST-SSRs for species identification. This marker set revealed significant genetic differentiation based on the fixation index and genetic distance values. Principal coordinate analysis and STRUCTURE analysis revealed distinct clusters of species and clearly identified red-sika hybrids. These markers showed applicability across different genera and proved suitable for identification and phylogenetic analyses across deer species.

## Introduction

After Bovidae, the family Cervidae is the most diverse of the order Artiodactyla/Cetartiodactyla. Fifty-five species in this group have been recognized, which are broadly distributed throughout the Northern Hemisphere [1]. Regrettably, half of these species are listed as

**Funding:** The research is funded by the Ministry of Science and Technology, Taiwan (102-2313-B-002-024-MY3, 108-2313-B-002-013-, 109-2313-B- 002-005-, 110-2313-B-002-053-) and Kenting National Park headquarters (489-103-01) to Dr. Yu-Ten Ju. The funders had no role in study design, data collection and analysis, decision to publish, or preparation of the manuscript.

**Competing interests:** The authors have declared that no competing interests exist.

vulnerable, endangered, critically endangered or extinct in the wild by the International Union for Conservation of Nature (IUCN). Several wild species are suffering increased risks due to anthropogenic interference [2]. For instance, human-mediated translocation has given rise to allopatric hybridization across different species, such as the interbreeding of introduced Japanese sika deer (*Cervus nippon*) to Scotland with indigenous red deer (*Cervus elaphus*) [3].

Deer across Asia, Europe, and Oceania are often managed as economically important animals that are farmed for their antler velvet and meat (venison) [4]. Species hybridization can enhance antler velvet growth via a heterotic effect. This practice has resulted in the allopatric hybridization of deer species worldwide. Red deer are commonly crossed with sika deer to improve antler growth. In New Zealand, wapiti (*Cervus canadensis*) have also been crossed with red deer, resulting in the introgression of wapiti genes into farmed red deer populations [5]. This process can harm the perceived quality and integrity of the farming industry [5,6]. As a traditional Chinese medicine, the market price for antler velvet can vary depending on the source species [7]. Premium pricing and production demands can promote the adulteration of antler velvet [8]. Thus, this industry is at risk of fraud resulting from procuring velvet from hybrid deer. Consequently, there is an evident need for better tracing of sources of velvet to validate product provenance. Significantly, fine-scale genetic analyses of deer populations would benefit both conservation research and deer farming in Asia, since sika deer and sambar occur as both farmed and wild species on the continent.

Detection technologies for identifying adulterated deer products have been developed using polymerase chain reaction (PCR)-based mitochondrial D-loop sequencing [7] to genotype antlers. However, mitochondrial sequencing analysis solely reveals the maternal lineage, so it is not suitable for detecting hybrids. Ultra-performance liquid chromatography quadrupole time-of-flight mass spectrometry (UPLC-QTOF-MS) based on enzyme-digested peptides has been applied to venison [9], and real-time polymerase chain reaction (RT–PCR) of the *kappa-casein precursor* gene has been employed to detect different deer species in the venison industry [10,11]. These technologies can only resolve species among specific samples, and they have not yet been proven capable of detecting hybrids.

Interspecific hybridization across members of Cervidae is common in both wild and domesticated populations, and such matings can produce fertile offspring, as observed in the crossbreeding of red deer with sika deer in Scotland [3]. The confident identification of hybrid deer using morphological traits alone is complicated. Their coloration and antler shape are not always intermediate between parental phenotypes [12]. For instance, in the case of red-sika hybridizations, hybrid deer display phenotypic variations related to carcass weight, jaw length, and incisor arcade breadth that reveal substantial additive genetic variation of these quantitative traits, hampering morphological-based identification [13]. Cross-genus hybridizations have also been reported, for example, between male sambar and female red deer [14]. The premise of hybrid detection is that markers should be able to clearly genotype species to reveal their ancestry. Previous published markers, such as species-diagnostic single nucleotide polymorphisms (SNPs), have been developed to detect hybridization events and genetic introgression only within a given genus [12,15]. Considering these studies together reveals a gap in our ability to diagnose source species and to identify cases of counterfeit deer products.

Simple sequence repeats (SSRs) are typically highly variable and, consequently, can be used for individual identification, paternity analysis, and pedigree construction, with applications in monitoring wildlife [16–18] and livestock breeding programs [19–21]. SSRs can be categorized into genomic-derived SSRs (gSSRs) or expressed sequence tag-derived SSRs (EST-SSRs). gSSRs display limited utility when performing cross-species amplification among different taxa [22,23]. Since EST-SSRs present better cross-species applicability, these markers have been developed for diverse animal taxa, including birds [22,24], domestic pigs (*Sus scrofa*)

[25], buffalo (*Bubalis bubalis*) [21], giant pandas (*Ailuropoda melanoleuca*) [26], and cattle (*Bos taurus*) [27]. EST-SSRs are derived from untranslated regions (UTRs) or coding regions (CDSs) in the genome, providing conserved flanking regions that enable primer pairs to be designed for genotyping [28,29]. Cross-species genotyping is critical for diagnosing fraudulent labeling and for identifying hybrid individuals. gSSRs have been used previously in studies of sika deer, red deer, and their hybrid offspring [3,13], but their effectiveness has only been assessed within the genus *Cervus*, and they have not been tested against different genera.

Transcriptome sequencing represents an effective approach for generating multiple EST sequences from nonmodel organisms [30,31]. This approach has been applied to sika deer [31,32], but EST-SSRs are still lacking for sambar, and no EST-SSR panel for deer has yet been subjected to cross-species amplification.

In this study, we aimed to develop a polymorphic EST-SSR panel for species identification across deer species from different genera. Based on the EST-derived character of this marker set, we hypothesized that it would show good transferability, allow reliable genotype interpretation, and could be employed to evaluate hybrid status among different members of the *Cervus* genus after capillary electrophoresis. Finally, we selected 16 polymorphic markers that revealed clear clustering across the three target deer species and identified two red-sika hybrids. Our EST-SSR panel can be used in cross-genus species identification and should prove helpful in wild deer conservation efforts, domestic breeding management, and the detection of product fraud.

## Materials & methods

### DNA sample collection

All animal experiments were approved by IACUC of National Taiwan University (Permit number: NTU107-EL-00234) and followed its Guidelines for Animal Experimentation.

Since red deer are not endemic to Taiwan and have been introduced mostly from New Zealand, we sampled red deer (n = 46) from commercial deer farms. An additional two sika-red deer hybrids (n = 2) were also sampled from commercial farms, and their ancestry was verified by the sequencing of mitochondrial DNA and the zinc finger Y-linked (ZFY) locus. We collected fresh antler velvet tissue slices when farmers were harvesting deer antlers. The deer were carefully restrained, and their antler velvet was cut with a saw. Slices of tissue were sampled with a sterile blade from the velvet tip of the antler and stored in 99% ethanol. Sambar samples (n = 45) were mostly collected from the Kaohsiung Animal Propagation Station, Pingdong, Taiwan (n = 30), and an additional 15 individuals were collected from deer farms. We obtained 30 sika deer samples from Shedding Nature Park, Pingdong, Taiwan. Blood samples from both sambar and sika deer were collected when the deer were undergoing veterinary health checks. EDTA-containing tubes were used for blood collection. DNA was extracted and purified from the antler velvet and blood samples using a Wizard Genomic Purification kit (Promega, WI) according to the manufacturer's procedure.

One sambar deer from a deer farm was selected to collect fresh antler velvet tissue for transcriptome sequencing. The tissue was stored in liquid nitrogen until RNA purification. Total RNA was extracted by using TRIzol (Invitrogen, CA) following the manufacturer's instructions. A total of 46.17 μg of RNA was collected for cDNA library construction.

### SSR mining and SSR marker development

Transcriptome sequencing from the velvet tip of the antler generated 67,054 unigenes. These unigenes were downloaded into MIcroSAtellite (MISA), software to identify microsatellites in the nucleotide sequences [33], and 11,190 SSRs were identified. Mononucleotides and short

SSRs (total length <15 base pairs (bp), i.e., SSRs with a minimum of 7 repetitions for dinucleotides, 4 repetitions for trinucleotides, 3 repetitions for tetranucleotides, 2 repetitions for pentanucleotides, or 2 repetitions for hexanucleotides) were excluded to retain SSRs with repeat motifs offering greater potential polymorphism. We obtained 2,179 SSRs that fit these criteria. Next, we culled SSRs that were allocated at the beginning or the end of a unigene because it was difficult to design primers for these loci in flanking regions. To avoid linkage disequilibrium, we ruled out SSRs allocated to the same unigene. In clusters containing unigenes with highly similar sequences (e.g., more than 70% sequence identity), the unigenes may come from the same gene or homologous genes. SSRs within the same cluster were also avoided. After screening based on the above criteria, we finally designed 103 primer pairs by using Primer 3–2.3.4 software [34,35].

## Detection of polymorphic SSRs

We conducted temperature gradient tests on 103 primer pairs to optimize annealing in PCR, and primer pairs that failed to produce specific PCR products were excluded. Fifty-five primer pairs were selected based on their clear and specific results in agarose gel electrophoresis. To screen for loci displaying high polymorphism, these 55 primer pairs were used for PCR in 45 sambar deer samples, and to determine their transferability, we conducted cross-species amplification in sika deer and red deer samples. The information of twenty-six polymorphic markers based on our sambar transcriptome sequences is listed in Table 1, and the allele frequencies of these markers in each species are shown in S1 Table.

PCR amplification was conducted using the Blend Taq Plus system (TOYOBO, Japan). We adopted a modified protocol involving reaction mixtures of 10 μL containing 6.5 μL of ddH$_2$O, 1.0 μL of 10X PCR buffer for Blend Taq, 1.0 μL of each dNTP (2.0 mM), 0.5 μL of the forward and reverse primers (10 μM), 1.0 μL of the DNA template (50 ng/μL), and 0.25 μL of Blend Taq Plus (2.5 U/μL). Reactions were conducted in an ABI PCR machine under the following conditions: 5 min at 94˚C, followed by 40 cycles of 30 sec at 94˚C, 30 sec at 58–61˚C, 30 sec at 72˚C, and a final elongation step of 10 min at 72˚C. The sizes of the PCR amplicons were measured via ABI 3730 capillary electrophoresis (Applied Biosystems, CA) at the National Center for Genome Medicine (NCGM), Taiwan.

## SSR sequence validation

To confirm the EST-SSR sequences of the amplicons, we sequenced all distinct alleles from each of the 29 polymorphic loci, and alleles that were only detected in heterozygous individuals were cloned by purifying the respective PCR products and introducing them into the pGEM-T easy vector (Promega, WI) before transforming them into competent DH10B cells. The clones with inserted EST-SSRs were selected and sequenced.

## Data analysis

The ABI 3730 outputs were read using Peak scanner version 1.0 software (Applied Biosystems, CA). MICROCHECKER [36] was applied to detect genotyping errors, including null alleles and allele dropout in each species. The number of alleles (NA), observed heterozygosity (Ho), expected heterozygosity (He), polymorphic information content (PIC), and probability of identity (PID) were calculated using Cervus version 3.0.3 [37]. The inbreeding coefficient ($F_{IS}$) was measured using Genetix v4.05 [38] with 10,000 permutations. The probability of exclusion (PE) was calculated in GenALEx 6.5 [39].

To achieve efficiency in species identification and phylogenetic analyses, we selected sixteen loci with a PIC>0.25, which were moderately informative (0.25<PIC<0.5) or highly

**Table 1. Primer information on 26 polymorphic microsatellite markers according to the sambar transcriptome sequences.**

| Locus | Primer sequence (5'->3') | Repeat unit | No. of repeat units | TA (˚C) | PCR product length | Fluorescent dye |
|---|---|---|---|---|---|---|
| Locus_3 | F: TCTCTGAAGAGACAGAGTCCTGC | TGC | 6 | 61 | 130 | 6-FAM |
| | R: AAAGAATGGCCCTCCCAAC | | | | | |
| Locus_4 | F: AGTTGCAGTTGAAGAAAGGACAG | GTG | 6 | 58 | 127 | 6-FAM |
| | R: GAATCAGTCAAACAAAGTGGGAG | | | | | |
| Locus_7 | F: CCTTTCAGGTCTCTCTGGAGG | GGA | 7 | 61 | 138 | PET |
| | R: AGCTGGCAAAGTCGGCTAC | | | | | |
| Locus_8 | F: GTACCCTAGAAATCCCACCTGAC | CAGA | 6 | 58 | 155 | PET |
| | R: ACTGCCGAGTCACTCAAAGG | | | | | |
| Locus_10 | F: GATGTATTCTCCCAGCCGTTAC | GCA | 7 | 58 | 136 | PET |
| | R: CTGATACATTGTGGTCTGCTGG | | | | | |
| Locus_14 | F: TGTCTCCCTTCTCTCATCTCATC | TC | 10 | 58 | 138 | 6-FAM |
| | R: CTTCCAAGCCAGGATATGTTATG | | | | | |
| Locus_15 | F: GCCATCTCTCCTCCCTTACTTAG | AC | 9 | 61 | 138 | PET |
| | R: GCAGAACCTTATCTGTTGGTGTC | | | | | |
| Locus_16 | F: AAGTCACTAAATCCTCCCTCCTG | TG | 9 | 61 | 141 | PET |
| | R: AACAACATGAGTGCTTATGCTCC | | | | | |
| Locus_20 | F: GTTCTCTGTCGTCTGGTGTGAG | GAT | 6 | 58 | 114 | 6-FAM |
| | R: AGAGTCGGACACGACTGAAGTG | | | | | |
| Locus_21 | F: AGATGACACTCAGGAGGATGGT | ACCCTG | 3 | 58 | 151 | PET |
| | R: CACATCCTATCCCAGGAGCTA | | | | | |
| Locus_25 | F: GAGCTCCTGAGGTTTACAGGTG | GACA | 5 | 58 | 147 | 6-FAM |
| | R: ACAGATGAGGAAACTGAGGTGTG | | | | | |
| Locus_26 | F: GTGCAGGAGGTGCTTGATGT | GCT | 6 | 58 | 105 | 6-FAM |
| | R: CAGCAGGAGAACAAGAGCAAC | | | | | |
| Locus_32 | F: ATCAACTGTGAGGATCAGCGTAG | TGTTT | 3 | 58 | 145 | 6-FAM |
| | R: TACCACTAAGTTATCCCTTGCCC | | | | | |
| Locus_34 | F: TATCAGCTAGTGAGTGGAAGC | TGG | 6 | 56 | 156 | VIC |
| | R: CTGTTCACAGCTTTGGTGTT | | | | | |
| Locus_37 | F: CTGTGACCATCTCTCCCTCCT | GTCTCC | 4 | 58 | 146 | 6-FAM |
| | R: GCAGTTTCTACCAGAGACCACAG | | | | | |
| Locus_39 | F: AGGGAACACAGCATGAAGATG | GAA | 6 | 58 | 145 | VIC |
| | R: CTTCAACTCTGACTGGCTTCTTT | | | | | |
| Locus_40 | F: AGCTTCCCAGTCTCTGACTTTCT | TC | 12 | 58 | 153 | VIC |
| | R: AGGATTTGGAGGGAGTGATATGT | | | | | |
| Locus_41 | F: GTAGTTTCTCCTTAGGCGTGGAT | TGAG | 5 | 58 | 135 | VIC |
| | R: CCACTGGAATCACAAAGTGTTCT | | | | | |
| Locus_42[a] | F: TGGCCTTTGATATGATACTGGAG | GTTT | 5 | 58 | 112 | VIC |
| | R: CGCACAACACATTATCTCAGAAC | | | | | |
| Locus_43 | F: CTTGCACTCTCAACCTACCTTGT | AC | 7 | 58 | 146 | VIC |
| | R: ACTCATTTCCAGAGCATCACAGT | | | | | |
| Locus_44[b] | F: TCAGTGACAATACACACTCGGTT | GT | 10 | 58 | 139 | VIC |
| | R: CCAGTTAACAGTGCAGATCCATT | | | | | |
| Locus_46 | F: CAGCACAGCAGATTCCCAG | CCTGC | 5 | 61 | 147 | 6-FAM |
| | R: TAAGTAAAGCAGCTGGGAGGAG | | | | | |
| Locus_48 | F: TTGTAACCAACACATAGCACACG | ACC | 4 | 58 | 144 | 6-FAM |
| | R: TCACCTCTGGGCTAATTGTAGAC | | | | | |

(*Continued*)

**Table 1.** (Continued)

| Locus | Primer sequence (5'->3') | Repeat unit | No. of repeat units | TA (˚C) | PCR product length | Fluorescent dye |
|---|---|---|---|---|---|---|
| Locus_49 | F: AGACCACATGTAAAACTGGCTGT | AC | 8 | 58 | 120 | 6-FAM |
| | R: CATACGTTTCTAGCCTGTTGCTT | | | | | |
| Locus_50 | F: ACCTATATGTTCTTCGGCTCCAT | GT | 9 | 58 | 137 | 6-FAM |
| | R: CTTTGGAACACTTGAGGAGACAT | | | | | |
| Locus_52 | F: GAACAACTGGATGCTGTG | GCC | 6 | 58 | 211 | 6-FAM |
| | R: GTTGAGTTGAGGCTGAGAAT | | | | | |
| Locus_53[b] | F: GTTGCAGGCCTTCTTTATC | TA | 10 | 58 | 162 | 6-FAM |
| | R: CAGATTCAAGGCTGTAGCA | | | | | |
| Locus_54 | F: GTGTTTCCTGAATCCAGATG | GCTGGG | 3 | 58 | 285 | 6-FAM |
| | R: GTGTTCTGTCCGTGCAAA | | | | | |
| Locus_55 | F: CTGGTTAACCTCTGAGAATCC | CCCCAT | 3 | 58 | 164 | 6-FAM |
| | R: GGAGTCAGAGTCACAGAGAAA | | | | | |

TA (˚C): Optimized annealing temperature; a: Failed to be amplified in sika deer; b: Failed to be amplified in red deer.

informative (PIC<0.5). We employed the genotyping results for these 16 loci in the following analyses (Table 1). The pairwise fixation index ($F_{ST}$) and the corresponding P value were calculated using Microsatellite Analyzer (MSA) version 4.5 [40] and Genepop 4.7 [41]. Principal coordinate analysis (PCoA) was conducted in GenALEx. We used STRUCTURE 2.3.4 [42] for assignment tests. This software models genetic structure by probabilistically assigning individuals to certain populations or to more than one population if the individuals are hybrids. The software generates estimates of the proportion of admixture (termed Q) for individuals in a sample set [42]. The test was performed with 10 iterations for each of three populations (K) with the Markov Chain Monte Carlo (MCMC) algorithm running for 500,000 generations, with an initial burn-in of 10,000 generations.

## Results

### *De novo* assembly of EST-SSRs from sambar antler velvet

We performed transcriptome sequencing to acquire EST-SSR sequences from an RNA sample taken from one sambar antler tip. The transcriptome assembly yielded 65,074 unigenes (mean length 1,131 bp), and 11,190 SSRs were identified by using MIcroSAtellite (MISA) software. After selection process described in the methods, primer pairs for PCR amplification were designed for the obtained set of 103 EST-SSR loci. We ruled out primer pairs that failed to yield specific PCR products from the sambar DNA template. Finally, 55 primer pairs were deemed suitable for further study and were labeled with fluorescent dyes to test their genotyping efficiency and polymorphism.

### Cross-genus amplification and characterization of polymorphic markers

We evaluated the polymorphism and cross-species transferability of the 55 candidate EST-SSRs across the sambar (n = 45), sika (n = 30), and red deer (n = 46) antler tip samples. A summary of the cross-species amplification of all 55 loci is shown in Table 2. The successful amplification rate was 94.5% (52/55) in sika deer and 89.1% (49/55) in red deer, and 21, 11 and 21 of the loci displayed polymorphism in sambar, sika deer and red deer, respectively. No signal of allele dropout or null alleles was detected among these polymorphic markers in any of the species with the sole exception of Locus_15, at which a null allele was detected in red

**Table 2. Cross-species amplification of 55 EST-SSR loci in the sambar, sika deer and red deer samples.**

| | No. of genotyped individuals | No. of successfully genotyped markers | No. of polymorphic loci | NA | Ho | He | PIC | $F_{IS}$ | PID | PE |
|---|---|---|---|---|---|---|---|---|---|---|
| Sambar | 45 | 55 | 21 | 2.81 | 0.3489 | 0.3523 | 0.3382 | 0.034 | 6.42E-09 | 0.998 |
| Sika deer | 30 | 52 | 11 | 2.31 | 0.4574 | 0.3855 | 0.3151 | -0.223 | 1.69E-05 | 0.970 |
| Red deer | 46 | 49 | 21 | 3.48 | 0.2804 | 0.2991 | 0.2761 | 0.063 | 6.82E-08 | 1 |

NA: Number of different alleles; Ho: Observed heterozygosity; He: Expected heterozygosity; PIC: Polymorphic information content; PID: Multilocus probability that two matching genotypes taken at random come from the same individual; PE: Probability of exclusion in parentage analysis.

deer. The average Ho and expected He ranged from 0.2804 to 0.4574 and 0.2991 to 0.3855, respectively, across the three deer species. The mean PIC of all three species was >0.25. PID ranged from 1.69E-05 to 6.42E-09, and PE ranged from 0.970 to 1 across the three species. The overall $F_{IS}$ ranged from -0.233 to 0.063 and showed no significant deviation from zero, indicating that our tested samples were not closely related. Among the 55 markers, three markers showed polymorphism in one species but produced nonspecific amplicons in other species. Specifically, Locus_42 was polymorphic in Sambar and red deer but failed to be amplified in sika deer. Locus 44 and Locus_53 were polymorphic in Sambar and sika deer but produced nonspecific results in red deer. Overall, a total of 29 EST-SSRs displayed polymorphism in all three deer species (see S1 and S2 Tables for details).

To increase efficiency and informativeness, we excluded markers with a PIC<0.25. Finally, 16 EST-SSR markers were chosen for phylogenetic analysis and were further characterized (Table 3). In this set of 16 markers, the number of alleles per locus ranged from 2 to 9, Ho ranged

**Table 3. Summary data of 16 polymorphic EST-SSRs in 45 sambar, 30 sika deer, and 46 red deer.**

| Locus | k | Ho | He | PIC | PID | $F_{ST}$ |
|---|---|---|---|---|---|---|
| Locus_7 | 6 | 0.588 | 0.708 | 0.667 | 0.126 | 0.211 |
| Locus_8 | 4 | 0.353 | 0.715 | 0.659 | 0.137 | 0.647 |
| Locus_14 | 6 | 0.184 | 0.578 | 0.486 | 0.270 | 0.760 |
| Locus_15 | 8 | 0.353 | 0.623 | 0.576 | 0.186 | 0.384 |
| Locus_16 | 5 | 0.252 | 0.387 | 0.336 | 0.438 | 0.327 |
| Locus_20 | 5 | 0.223 | 0.329 | 0.307 | 0.468 | 0.155 |
| Locus_25 | 3 | 0.198 | 0.452 | 0.384 | 0.370 | 0.698 |
| Locus_26 | 3 | 0.008 | 0.381 | 0.310 | 0.449 | 0.986 |
| Locus_34 | 4 | 0.421 | 0.652 | 0.576 | 0.194 | 0.645 |
| Locus_40 | 5 | 0.504 | 0.675 | 0.612 | 0.171 | 0.443 |
| Locus_41 | 8 | 0.292 | 0.426 | 0.410 | 0.343 | 0.261 |
| Locus_43 | 6 | 0.689 | 0.769 | 0.729 | 0.092 | 0.162 |
| Locus_46 | 2 | 0.008 | 0.476 | 0.362 | 0.389 | 0.988 |
| Locus_49 | 2 | 0.269 | 0.333 | 0.277 | 0.494 | 0.262 |
| Locus_50 | 9 | 0.271 | 0.748 | 0.708 | 0.102 | 0.678 |
| Locus_52 | 2 | 0.109 | 0.413 | 0.327 | 0.432 | 0.742 |
| Mean | 4.9 | 0.2951 | 0.5416 | 0.4829 | | 0.5464 |
| Total | | | | | 2.62E-10 | |

Ho: Observed heterozygosity; He: Expected heterozygosity; PIC: Polymorphic information content; PID: Multilocus probability that two matching genotypes taken at random come from the same individual; $F_{ST}$: Fxation index.

*De novo* assembly of
sambar transcriptome

↓ MISA microsatellite mining

11,190 SSR loci in
65,074 unigenes

↓ Primer3 primer design

103 loci and designed
primer pairs for genotyping

↓ PCR amplification test

55 loci with high
specificity in sambar

↓ Cross-genus genotyping

29 polymorphic loci detected
among sambar, sika and red

↓ Picking out loci with
low PIC value (PIC<0.25)

16 polymorphic loci selected
for species identification

↓ Data analysis

Testing the power of
species identification
by 16 markers

**Fig 1. Flowchart of marker development for 16 cross-genus EST-SSRs.**

from 0.008 to 0.689 (mean = 0.2951), and He ranged from 0.329 to 0.769 (mean = 0.5416). Although only one individual showed heterozygosity at Locus_26 and Locus_46, they still revealed a reasonable PIC across the 121 deer samples because different alleles had become fixed in different deer species. For the 16-marker set, PIC ranged from 0.307 to 0.729 (mean = 0.4829), and PID ranged from 0.102 to 0.494 (Combined = 2.62E-10). A summary flow-chart of the process of marker selection for species identification is shown in Fig 1.

## Evaluation of the genetic differentiation capability of the 16-marker set for species identification

We used the set of 16 EST-SSR markers to determine its species identification power across sambar, sika deer and red deer. For this purpose, we calculated genetic distance (Nei's $D_A$) and

**Table 4. Nei's DA (below the diagonal) and pairwise $F_{ST}$ (above the diagonal) for 16 EST-SSR markers among the three deer species.**

| Species | red deer | sika deer | sambar |
|---|---|---|---|
| Red deer | | 0.4148 | 0.6472 |
| Sika deer | 0.2663 | | 0.5659 |
| Sambar | 0.4858 | 0.5806 | |

$F_{ST}$ using 10,000 permutations of allele frequencies (Table 4). We observed a shorter Nei's DA (0.2663) between sika and red deer than between sika deer and sambar (0.5806) or red deer and sambar (0.4858). The pairwise $F_{ST}$ results mirrored those of Nei's DA, with the $F_{ST}$ value between red and sika deer (0.4148) being lower than that between red deer and sambar (0.6472) or sika and sambar (0.5260). Pairwise $F_{ST}$ analysis across all samples revealed significant differentiation (P value<0.001) among the three deer species.

To better illustrate genetic separation among the three species, we conducted PCoA in GenALEx (Fig 2). Two known red-sika hybrid deer were used as controls. Our results show that the three species were clearly distinguishable, with explanatory scores for Coordinate 1 and Coordinate 2 of 44.07% and 15.13%, respectively. As anticipated, the two known hybrid individuals, HY01 and HY02 (purple circles), were positioned between the sika and red deer clusters.

To analyze the genetic structure of 123 deer individuals, we performed assignment testing in STRUCTURE software. All but two individual samples were unambiguously assigned to one of three clusters (K = 3), which represented sika deer (blue cluster), red deer (red cluster), and sambar deer (green cluster) (Fig 3). The two deviant samples, representing the hybrid individuals HY01 and HY02, were split between the sika deer and red deer clusters as expected. For HY01, the Q value was 0.459 for sika deer and 0.539 for red deer. For HY02, the Q value was 0.331 for sika deer and 0.664 for red deer.

## Discussion

In the present study, we used *de novo* assembly to establish a panel of EST-SSR loci to confidently identify species of Cervidae. Our main goal was to develop a set of genetically based markers allowing the identification of deer species. In recent years, many studies have focused on comparing the utility of microsatellite SSRs and SNPs. Ross et al. [43] estimated relatedness

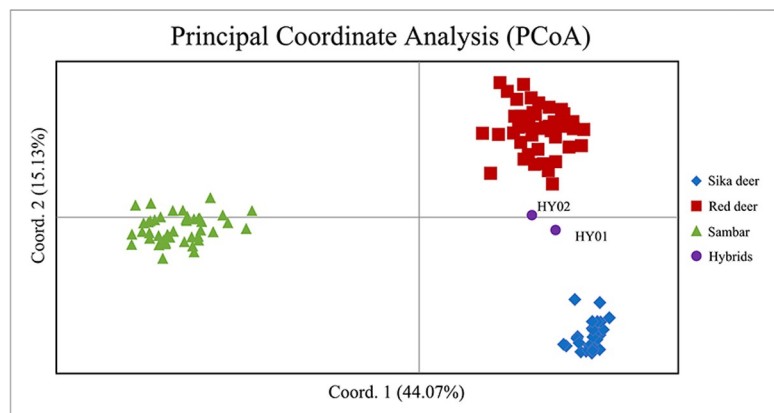

**Fig 2. Principal coordinate analysis of three deer species and two hybrids (HY01 and HY02) using the 16-marker set.**

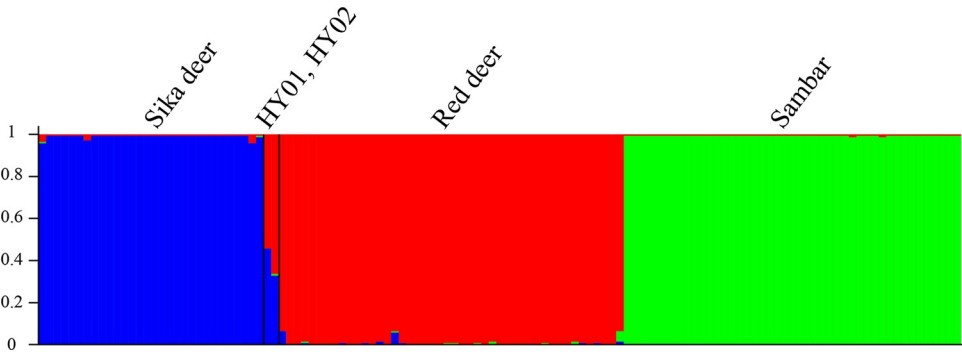

**Fig 3. Assignment test in STRUCTURE for 123 deer samples based on 16 EST-SSR markers.**

in Chinese rhesus macaques (*Macaca mulatta*) and suggested that SSRs offer more precise predictive power than SNPs for establishing how individuals are related. Fernández et al. [44] examined the effectiveness of SSRs and SNPs in a consanguineous Angus cattle herd (*Bos taurus*) and found that twice as many SNP markers as SSRs were required to achieve the same effectiveness in individual identification and parentage analysis. Sorkheh et al. [45] similarly showed that SSRs present higher PIC than SNPs. Furthermore, SSR markers have been successfully used to genotype samples with low DNA quality, exhibiting greater efficacy in such samples than SNPs and genotyping-by-sequencing methods [22]. SSRs have been successfully and widely applied for cross-species amplification in diverse vertebrates, including cetaceans, birds, and frogs [46]. Mitrus et al. [47] used gSSR markers developed from *Ficedula hypoleuca* and found that 13 of the 24 tested primer pairs (54%) could be used in *Ficedula parva*. Tardy et al. [48] assessed the amplification of 32 markers derived from *Balaenoptera physalus* in four additional cetacean species, revealing transferability >72%. Maduna et al. [49] tested the cross-amplification of 11 gSSRs across six shark species, demonstrating genotyping success rates of 72%-100%. However, SSRs developed in certain taxa, such as fishes [50], display a low probability of cross-amplification or shared polymorphism. To the best of our knowledge, no other assessment of EST-SSR marker cross-amplification has been reported in Cervidae. Our novel panel of 16 EST-SSRs shows high transferability and provides a reasonably high information content (0.5>PIC>0.25 and PIC>0.25, Table 2) [51]. Markers developed from ESTs show slightly lower polymorphism [31,52]. While markers with a reasonably high PIC are less informative, they can be combined with other high-information-content markers for phylogenetic analysis [53], as reported in the unweighted pair group method with arithmetic mean (UPGMA) analysis of domestic breeds [54] and the analysis of molecular variance (AMOVA) and discriminant analysis of principal components (DAPC) of papaya [55]. In addition to their phylogenetic applicability, we selected these reasonably high-PIC markers because they showed private alleles or the fixation of a certain allele in one of the species (S1 Table). For example, Locus_26 was fixed in sika deer, with a private allele size = 97 base pairs, while its PIC value was 0.310 across the three species. This marker played an important role in indicating the sika deer origin of a species-unknown individual. To include these markers in our species identification marker set, we set the criterion of 0.25<PIC<0.50, and 16 markers were retained for further analysis.

The wild sika deer population in Kenting National Park, Taiwan, is derived from a small population comprising 5 males and 17 females that were reintroduced to the park from the Taipei Zoo in 1994. Thus, this founder event may explain the low polymorphism displayed by our sika samples. In contrast, the high polymorphism observed in our red deer samples may

be due to the diverse origins of farmed red deer. Furthermore, farmed red deer are often mated with wapiti, resulting in higher genetic diversity in red deer populations. The combined PID values were 6.42E-9, 1.69E-5, and 2.90E-7 for our sambar, sika and red deer sample sets, respectively, demonstrating that our EST-SSR panel competently achieved individual identification, requiring a PID<0.01 [56,57]. This panel also displayed utility for pedigree establishment in parentage analysis, with mean PE values ranging from 97.0% - 99.9%. Accordingly, our marker set should assist in the breeding management of farmed deer.

The refined set of 16 polymorphic markers revealed the lowest pairwise genetic distance (mean Nei's DA = 0.2663) and pairwise $F_{ST}$ (mean = 0.4148) between red and sika deer samples, corroborating a previous mitochondrial DNA study [58] and the current taxonomy [59,60]. $F_{ST}$ values revealed that this set of markers could distinguish all three tested species. PCoA validated that outcome, showing that, apart from the two hybrid samples, all individuals were divided into three distinct clusters corresponding to the three species (Fig 3). The first two axes of our PCoA accounted for 59.2% of the variation in the dataset, which is higher than the percentages of variation explained by markers developed for species identification in other studies, such as studies of zebra [61] and Elasmobranchii [62]. Assignment analysis in STRUCTURE showed that all 123 samples could be sorted into the three *a priori* clusters (K = 3), with HY01 and HY02 again showing admixed ancestry. Based on these results, we propose that our panel of 16 EST-SSRs can be used to accurately discriminate the three deer species from the *Cervus* and *Rusa* genera, which should prove useful in detecting counterfeit deer products and in breeding programs.

## Conclusions

We successfully performed the cross-species amplification of 55 EST-SSR loci in three related species belonging to two genera in the family Cervidae. We selected a panel of 16 EST-SSRs that displayed unambiguous genotyping and informativeness for population genetics and phylogenetic analyses. This panel can be used for species identification and hybrid detection across deer genera.

## Supporting information

**S1 Table. Allele frequency of 29 EST-SSR markers among sambar, sika deer, and red deer.** (XLSX)

**S2 Table. Marker polymorphism information of 29 EST-SSRs among sambar, sika deer and red deer.** k, number of alleles; PIC, polymorphism information content; P(ID), HWE, deviation from Hardy-Weinberg equilibrium (NS not significant, ND not done, *P<0.05). LD, linkage disequilibrium. Microchecker null, MICROCHECKER results of tests for evidence of null allele. Microchecker dropout, MICROCHEKER results of tests for evidence of allele dropout. (XLSX)

**S3 Table. Pairwise Queller and Goodnight relatedness (above the diagonal) and relatedness based on maximum likelihood method (below the diagonal) of each individual in sambar, red deer, and sika deer by polymorphic EST-SSRs.** Pairwise Queller and Goodnight relatedness was calculated by GenALEx 6.5. Relatedness based on maximum likelihood method was calculated by ML-Relate (https://www.montana.edu/kalinowski/software/ml-relate/index.html). (XLSX)

## Acknowledgments

We thank the National Center for Genome Medicine for technical support with genotyping. We thank Taroko National Park headquarters and Kenting National Park headquarters for deer sample collection. We are grateful to Mukesh Thakur and two anonymous reviewers for their valuable suggestions. We thank John O'Brien for the English correction. We would also like to acknowledge the professional manuscript services of American Journal Experts.

## Author Contributions

**Conceptualization:** Chien-Yi Hung, Pei-Yu Sun, Yu-Ten Ju.

**Data curation:** Chen Hsiao, Hsin-Hung Lin, Shann-Ren Kang, Chien-Yi Hung, Chieh-Cheng Yu, Kok-Lin Toh.

**Formal analysis:** Chen Hsiao, Pei-Ju Yu.

**Funding acquisition:** Yu-Ten Ju.

**Investigation:** Chen Hsiao, Pei-Yu Sun, Chieh-Cheng Yu.

**Methodology:** Chen Hsiao, Chien-Yi Hung.

**Resources:** Hsin-Hung Lin, Shann-Ren Kang.

**Software:** Chen Hsiao.

**Supervision:** Yu-Ten Ju.

**Validation:** Chen Hsiao, Pei-Yu Sun, Kok-Lin Toh, Pei-Ju Yu.

**Writing – original draft:** Chen Hsiao.

**Writing – review & editing:** Chen Hsiao, Yu-Ten Ju.

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
