## [Decision Letter · Decision Letter 0]

12 Nov 2021

PONE-D-21-07853

Development of 16 novel EST-SSR markers for species identification and cross-genus amplification in sambar, sika, and red deer

PLOS ONE

Dear Dr. Ju,

Thank you for submitting your manuscript to PLOS ONE. After careful consideration, we feel that it has merit but does not fully meet PLOS ONE’s publication criteria as it currently stands. Therefore, we invite you to submit a revised version of the manuscript that addresses the points raised during the review process.

We look forward to receiving your revised manuscript.

Kind regards,

Mukesh Thakur, Ph.D.

Academic Editor

PLOS ONE

Additional Editor Comments (if provided):

I now checked the manuscript, entitled "De novo characterization of the sambar antler velvet transcriptome and EST-SSR development for cross-genus species-identification" . I found it significant as the developed new EST-SSR markers can be applicable for population genetic studies of deer species. Based on two reviewer's comments and my own evaluation of the contents, I would recommend manuscript as "Major Revision" and would advise reviewer to estimate the genotyping errors for the markers prioritized as the significant occurrence of allele drop out, null alleles, false alleles may affect the overall explanation/outcome of the results. I suggest authors to consider the reviewer's comments and revise the manuscript accordingly.

Journal Requirements:

A clean copy of the edited manuscript (uploaded as the new *manuscript* file).

3. In your Methods, please describe the exact protocol used for the collection of the antler velvet samples.

4. PLOS requires an ORCID iD for the corresponding author in Editorial Manager on papers submitted after December 6th, 2016. Please ensure that you have an ORCID iD and that it is validated in Editorial Manager. To do this, go to ‘Update my Information’ (in the upper left-hand corner of the main menu), and click on the Fetch/Validate link next to the ORCID field. This will take you to the ORCID site and allow you to create a new iD or authenticate a pre-existing iD in Editorial Manager. Please see the following video for instructions on linking an ORCID iD to your Editorial Manager account: https://www.youtube.com/watch?v=_xcclfuvtxQ.

Reviewers' comments:

Reviewer's Responses to Questions

**Comments to the Author**

1. Is the manuscript technically sound, and do the data support the conclusions?

Reviewer #1: Yes

Reviewer #2: Yes

2. Has the statistical analysis been performed appropriately and rigorously? 

Reviewer #1: Yes

Reviewer #2: Yes

3. Have the authors made all data underlying the findings in their manuscript fully available?

Reviewer #1: No

Reviewer #2: Yes

4. Is the manuscript presented in an intelligible fashion and written in standard English?

Reviewer #1: Yes

Reviewer #2: Yes

5. Review Comments to the Author

Reviewer #1: Comments:

The manuscript “Development of 16 novel EST-SSR markers for species identification and cross-genus amplification in sambar, sika, and red deer.” by Hsiao et al. developed new EST-SSR markers for genetic identification of deer species. These markers demonstrate applicability across different genera and proved suitable for identification and phylogenetic analyses across deer species. The manuscripts is well written and developed SSR markers to resolve different issue related to the deer species conservation. Although the use of the methodology and analytical approaches seem appropriate, but there are some flaws in presentation of results and it is advised to authors to incorporate suggestions:

1. Give reference in line Number 51.

2. Among the 16 SSR markers selected for the species identifications, eight markers showed lower PIC (polymorphism Information Content) value i.e. <0.5, which is not acceptable for Species, Individual and population Genetic structure Identification. Lower PIC value may reduce the discriminatory power of theses markers may reduce. Authors need to Clarify that why they are selecting these markers.

3. Authors did nice effort to screen SSR markers and demonstrated their use in the species identification but I could not see any effort to estimate the genotyping error. Genotyping error is inherent problem with SSR markers. Especially with degraded samples (like Antlers, Hair and Fecal samples). Presence of genotyping error (allele dropout and null allele) may mislead the shorting of population/species based on the allele frequency clustering. Therefore, authors need to address the Genotyping error issue for all the 16 selected markers.

4. Relatedness assessment is need to justify that the tested samples did not include only closely related individuals are not present in the data.

Reviewer #2: Reviewer comment:

In recent times, the population status of many deer species are declining with few species on the verge of extinction. Most of the published population genetic and hybridization studies on deer species have used cross species amplification microsatellites. The current study entitled "De novo characterization of the sambar antler velvet transcriptome and EST-SSR development for cross-genus species-identification" tries to fill the lacunae by designing microsatellites for Sambar and tested their efficacy in other two deer species.

The study holds significance. The authors tried to answer majority of the earlier reviewer’s comments. But still few sections are confusing and require changes to be made especially the marker designing, selection and annotation sections for better understanding for readers. Moreover, I suggest the authors to remove transcriptome and annotation part in the results and discussion sections as it has no significance in this study.

Few suggestions and questions :

Introduction: Hypothesis is not well established in the last paragraph of Introduction.

Line 80-81: Give citation for the sentence

Line 156-162: How many SSR’s were retained finally for primer designing?

Line 163-165: Elaborate the methodology a bit by including the wet lab work and any other software used in selecting 55 primers from a total of thousands of SSR’s

Line 168-169: I believe “ Information on twenty-six polymorphic markers based

on our sambar transcriptome sequence is listed in Table 1” should be moved to results. Also there is no proper justification for reporting only 26 primers out of 55 in table 1.

Line 170: Report if you have information on chromosome number in Table-1 from which the loci is picked. I believe the chromosome number is important in population genetic analysis to avoid any linkage disequilibrium.

Line 187-188: Reason for selecting 16 primers out of 55 selected and 26 in Table 1 is missing. It’s too confusing, please clarify wherever you are reducing the number of microsatellites.

Line 194: SSR sequence validation- I believe this section should go before finalizing the SSR’s for further analysis. Also mention how many microsatellites were used for validation.

Line 215-236: Though most of the above questions were answered here in these paragraphs, it’s too confusing and I believe it should be in methodology section for better understanding for readers.

Line 237: Report results of HWE and LD tests too for each species in Table 2 as they are crucial for population genetic analysis.

Line 237-238: I suggest you to merge both Tables 2 and 3 and report single table with basic diversity estimates for all three species.

Line 253-254: Overall and individual species wise PIC values were very less. Can you comment on the reliability of selected 16 markers for its use in other population genetic and hybridization studies?

Line 272: Remove asterisk marks on the FST p-values as all are significant.

Line 282-286: Move to methodology section.

Line 351: Replace ‘they’ with ‘the’.

Line 426-427: Mention population genetics and hybridization studies in the final sentence of conclusion section.

S1 Table: I suggest S1 table should be moved to methodology section as it more clear than writing part.

6. PLOS authors have the option to publish the peer review history of their article (what does this mean?). If published, this will include your full peer review and any attached files.

Reviewer #1: No

Reviewer #2: No

---

## [Author Response · Author response to Decision Letter 0]

18 Jan 2022

Revise to reviewers

Additional Editor Comments:

Response:

We have checked the format. We corrected the symbol of the corresponding author. We modified figures by PACE, according to journal requirements.

A clean copy of the edited manuscript (uploaded as the new *manuscript* file).

Response:

We have asked the edition service from AJE. We acknowledged the editor of AJE and another English editor for their English correction in Acknowledgements. We also prepared a supporting information file named “manuscript with track change by AJE”, This file only showed track changes by AJE, comparing to “Revised Manuscript with Track Changes”, which remaining all changes during the revision.

3. In your Methods, please describe the exact protocol used for the collection of the antler velvet samples.

Response:

We described the protocol used for antler velvet collection at Line 139-143.

4. PLOS requires an ORCID iD for the corresponding author in Editorial Manager on papers submitted after December 6th, 2016. Please ensure that you have an ORCID iD and that it is validated in Editorial Manager. To do this, go to ‘Update my Information’ (in the upper left-hand corner of the main menu), and click on the Fetch/Validate link next to the ORCID field. This will take you to the ORCID site and allow you to create a new iD or authenticate a pre-existing iD in Editorial Manager. Please see the following video for instructions on linking an ORCID iD to your Editorial Manager account: https://www.youtube.com/watch?v=_xcclfuvtxQ.

Response:

We renewed the ORCID information of the corresponding author Yu-Ten Ju.

Reviewer1

1. Give reference in line Number 51.

Changes made:

We added the reference at Line 53. Hoffmann et al. (2015) stated that “Overexploitation of natural populations (illegal hunting, etc.) and habitat loss have led to the severe population reductions”. According to this statement and to prevent over-interpret the description from reference, we removed the wordings of “the risks of extinction”. 

2. Among the 16 SSR markers selected for the species identifications, eight markers showed lower PIC (polymorphism Information Content) value i.e. <0.5, which is not acceptable for Species, Individual and population Genetic structure Identification. Lower PIC value may reduce the discriminatory power of theses markers may reduce. Authors need to Clarify that why they are selecting these markers.

Changes made:

Thanks to reviewer’s insightful comments, we put forward our explanation based on the reviewer’s comments, so that our article can be more clear and reasonable. We agree that PIC>0.5 would be a better choice for phylogenetic analysis. However, the relatively low level of polymorphic information content in EST-SSR markers may be due to the source sequence of these markers. EST are more conserved sequences comparing to genomic sequences, which are spread throughout the genome, and markers developed from EST would feature slightly lower polymorphism (Khimoun et al. 2017; Jia et al. 2019). In addition, markers with 0.25<PIC<0.5 were classified as “reasonably informative” by Botstein et al. (1980) and were used in several phylogenetic analysis, such as horse breeds for UPGMA (Zeng et al., 2018) or in papaya for AMOVA (Matos et al., 2013), which including reasonably informative markers when conducting analysis. Moreover, some markers with strong power in species discrimination but did not show high PIC value. For example, while Locus_26 featuring PIC=0.310, it was fixed in sika and would be a good indicator for distinguishing red and sika. To include those markers in our analysis, we set our criteria at 0.25<PIC. In order to illustrate the rationality of using these reasonably informative EST-SSR markers, we have added explanations and supporting references in the Discussion section (Line 339-355).

3. Authors did nice effort to screen SSR markers and demonstrated their use in the species identification but I could not see any effort to estimate the genotyping error. Genotyping error is inherent problem with SSR markers. Especially with degraded samples (like Antlers, Hair and Fecal samples). Presence of genotyping error (allele dropout and null allele) may mislead the shorting of population/species based on the allele frequency clustering. Therefore, authors need to address the Genotyping error issue for all the 16 selected markers.

Changes made:

The reviewer did point out the weakness of our original manuscript in verifying the EST-SSR markers. According reviewer’s suggestion, we checked allele dropout and null allele in each species by MICROCHECKER and the result clearly showed no genotyping error was found. In null allele detection, we detected only one null allele i.e. Locus_15 at red deer. We added these descriptions of MICROCHECKER in Methods (Line 207, 208) and Results (Line 246-248).

4. Relatedness assessment is need to justify that the tested samples did not include only closely related individuals are not present in the data. 

Changes made:

We added overall FIS of each species at Table 2. Data analysis methods were described in Methods (Line 211,212). FIS value and significant tests were described in Results (Line 252-254). We stated that “no significant deviation from zero, indicating that our tested samples were not closely related.” at Line 252-254. 

Reviewer2

In recent times, the population status of many deer species are declining with few species on the verge of extinction. Most of the published population genetic and hybridization studies on deer species have used cross species amplification microsatellites. The current study entitled "De novo characterization of the sambar antler velvet transcriptome and EST-SSR development for cross-genus species-identification" tries to fill the lacunae by designing microsatellites for Sambar and tested their efficacy in other two deer species.

The study holds significance. The authors tried to answer majority of the earlier reviewer’s comments. But still few sections are confusing and require changes to be made especially the marker designing, selection and annotation sections for better understanding for readers. Moreover, I suggest the authors to remove transcriptome and annotation part in the results and discussion sections as it has no significance in this study.

Changes made:

We removed paragraph of unigene annotation of SSR loci from Result and Discussion sections. While partial description of transcriptome assembly was remained because we want to explain how we got these SSRs.

Few suggestions and questions :

Introduction: Hypothesis is not well established in the last paragraph of Introduction.

Changes made:

We put forward the hypothesis of this manuscript based on the reviewer to make the topic more clear. We proposed a hypothesis as following and wrote down at Line 120-123 in the section of Introduction: 

“Based on the EST-derived character of this marker set, we hypothesized that it would show good transferability, allow reliable genotype interpretation, and could be employed to evaluate hybrid status among different members of the Cervus genus after capillary electrophoresis.”

Line 80-81: Give citation for the sentence

Changes made:

We gave the example of the hybridization of red deer and sika deer and cited the reference (Line 85). 

Line 156-162: How many SSR’s were retained finally for primer designing?

Changes made:

We got 103 SSRs after the screening. We added the number of the retained SSRs in the Methods section (Line 170-172).

Line 163-165: Elaborate the methodology a bit by including the wet lab work and any other software used in selecting 55 primers from a total of thousands of SSR’s

Changes made:

We described the criteria in selecting proper SSR loci in detail (Line 160-177) and works on wet lab examination to select the 55 primers (Line 176, 177). 

Line 168-169: I believe “ Information on twenty-six polymorphic markers based

on our sambar transcriptome sequence is listed in Table 1” should be moved to results. Also there is no proper justification for reporting only 26 primers out of 55 in table 1

Changes made:

Following reviewer’s later suggestion, we moved Table S1 to Methods. Since Table S1 has mentioned some marker information such as locus name, we think remaining Table 1 in Methods would make the manuscript more understandable. Hence, we decide to remain Table 1 in Methods. Besides, we explained the reason why we only showed 29 primer pairs here that, we selected 29 primers because their PIC value > 0.25. This description was added at Line 267-269. We increase three primer pairs here to make the manuscript more logical. These three primers, i.e. Locus_42, Locus_44, and Locus_53, while showed polymorphism, were not shown in the original manuscript because they failed to be amplified in one of the three species (see Table S2 and Line 254-259). However, to recall the number of polymorphic markers we showed in Table 2, we added these markers in Table 1 and noted that they could not be amplified in all three species and also added description at Line 254-259. For the above reasons, the number of primer changed from 26 to 29. Allele frequency for all markers in each species (Table S1) was also modified to 29 markers.

Line 170: Report if you have information on chromosome number in Table-1 from which the loci is picked. I believe the chromosome number is important in population genetic analysis to avoid any linkage disequilibrium.

Changes made:

No sambar whole genome sequencing with chromosome construction was available online. Hence, we used red deer whole genome chromosome assembly (Cervus elaphus mCerEla1.1; Bana et al., 2018) to refer the chromosome number of each locus. Chromosome number of each locus was showed at Table S2.

Bana NÁ, Nyiri A, Nagy J, Frank K, Nagy T, Stéger V, Schiller M, Lakatos P, Sugár L, Horn P, Barta E, Orosz L. The red deer Cervus elaphus genome CerEla1.0: sequencing, annotating, genes, and chromosomes. Mol Genet Genomics. 2018 Jun;293(3):665-684. doi: 10.1007/s00438-017-1412-3. Epub 2018 Jan 2. PMID: 29294181.

Line 187-188: Reason for selecting 16 primers out of 55 selected and 26 in Table 1 is missing. It’s too confusing, please clarify wherever you are reducing the number of microsatellites.

Changes made:

Thank reviewer for their suggestion. Our expression is not clear in the original manuscript. The criteria of how we selected the sixteen loci was PIC>0.25 . Regarding the selection criteria and details of these 16 pairs of primer pairs, we added these descriptions in Line 267-269.

Line 194: SSR sequence validation- I believe this section should go before finalizing the SSR’s for further analysis. Also mention how many microsatellites were used for validation.

Changes made:

Thanks reviewer's suggestion, we moved the paragraph of SSR sequence validation to data analysis according to the reviewer's suggestion to make the manuscript more fluent(Line 198-204), and we also added the number of microsatellites that we were used for validation at Line 200. 

Line 215-236: Though most of the above questions were answered here in these paragraphs, it’s too confusing and I believe it should be in methodology section for better understanding for readers.

Changes made:

We reorganizes the confusing parts of our texts and we revised the context to make it more logical and avoid confusion. We added descriptions of details of how we selected 103 primer pairs (Line 164-172) and 55 primer pairs (Line 174-177). Criteria of the selection of 16 cross-genus markers were described at Line 214-216 in Methods, Line 267-268 in Results, and Line 339-341 in Discussion. Besides, we followed reviewer’s suggestion that we removed some descriptions about the transcriptome sequencing, and we rewrote the section of “De novo assembly of EST-SSRs from sambar antler velvet” and only remained the passage about unigene generation derived from transcriptome sequencing.

Line 237: Report results of HWE and LD tests too for each species in Table 2 as they are crucial for population genetic analysis.

Changes made:

We provided a new supplemental table (Table S2) for describing marker information for each species, including HWE, LD, and the number of potential chromosome location based on red deer genome. 

Line 237-238: I suggest you to merge both Tables 2 and 3 and report single table with basic diversity estimates for all three species.

Changes made:

We remained the two tables because these tables represented different information of the marker set. Table 2 showed the polymorphism of markers in each species. This result showed information of transferability of markers in each species. Besides, NA, heterozygosity, and PIC represented the efficiency of our panel for selecting polymorphic markers from EST. In contrast, Table 3 showed polymorphism and FST of each marker, which provided information for users to decide which markers they would like to choose. According to these reasons, we decided to remain the two tables.

Line 253-254: Overall and individual species wise PIC values were very less. Can you comment on the reliability of selected 16 markers for its use in other population genetic and hybridization studies?

Changes made:

We added discussions (Line 339-355) to explain why we set a relatively low PIC value (>0.25) as our threshold for cross-genus genotyping. The explanation was as follow:

“Our novel panel of 16 EST-SSRs shows high transferability and provides a reasonably high information content (0.5>PIC>0.25 and PIC>0.5, Table 3) [51]. Markers developed from ESTs show slightly lower polymorphism [31, 52]. While markers with a reasonably-informative PIC are less informative, they can be combined with other high-information-content markers for phylogenetic analysis [53], as reported in the unweighted pair group method with arithmetic mean (UPGMA) analysis of domestic horse breeds [54] and the analysis of molecular variance (AMOVA) and discriminant analysis of principal components (DAPC) of papaya [55]. In addition to their phylogenetic applicability, we selected these reasonably informative PIC markers because they showed private alleles or the fixation of a certain allele in one of the species (S1 Table). For example, Locus_26 was fixed in sika deer, with a private allele size = 97 base pairs, while its PIC value was 0.310 across the three species. This marker played an important role in indicating the sika deer origin of a species- unknown individual. To include these markers in our species identification marker set, we set the criterion of PIC>0.25, and 16 markers were retained for further analysis.” 

 Also, we removed the statement of “highly informative” of our marker set.

Line 272: Remove asterisk marks on the FST p-values as all are significant.

Changes made:

We removed asterisk marks in Table 2. 

Line 282-286: Move to methodology section.

Changes made:

We moved the passage of STRUCTURE methodology to Data analysis at Line 220-224 in Methods section.

Line 351: Replace ‘they’ with ‘the’.

Changes made:

We revised the wording at Line 359 “the high polymorphism”.

Line 426-427: Mention population genetics and hybridization studies in the final sentence of conclusion section.

Changes made:

We highlighted the power of population genetics and hybridization for our marker set at Line 389.

S1 Table: I suggest S1 table should be moved to methodology section as it more clear than writing part.

Changes made:

We agree reviewer’s opinion. We referred S1 Table at Line 182 in Methods.

---

## [Decision Letter · Decision Letter 1]

1 Mar 2022

Development of 16 novel EST-SSR markers for species identification and cross-genus amplification in sambar, sika, and red deer

PONE-D-21-07853R1

Dear Sir/Madam,

We’re pleased to inform you that your manuscript has been judged scientifically suitable for publication and will be formally accepted for publication once it meets all outstanding technical requirements.

Kind regards,

Mukesh Thakur, Ph.D.

Academic Editor

PLOS ONE

Additional Editor Comments (optional):

Pls include estimates of genotyping error and relatedness as supplementary file before processing the MS for formal acceptance.

Reviewers' comments:

Reviewer's Responses to Questions

**Comments to the Author**

1. If the authors have adequately addressed your comments raised in a previous round of review and you feel that this manuscript is now acceptable for publication, you may indicate that here to bypass the “Comments to the Author” section, enter your conflict of interest statement in the “Confidential to Editor” section, and submit your "Accept" recommendation.

Reviewer #1: All comments have been addressed

Reviewer #2: All comments have been addressed

2. Is the manuscript technically sound, and do the data support the conclusions?

Reviewer #1: Partly

Reviewer #2: Yes

3. Has the statistical analysis been performed appropriately and rigorously? 

Reviewer #1: Yes

Reviewer #2: Yes

4. Have the authors made all data underlying the findings in their manuscript fully available?

Reviewer #1: Yes

Reviewer #2: Yes

5. Is the manuscript presented in an intelligible fashion and written in standard English?

Reviewer #1: Yes

Reviewer #2: Yes

6. Review Comments to the Author

Reviewer #1: In the revised manuscript authors responded to all the questions except questions 3 and 4. Therefore I still have a concern about these two comments given below:

Comments 1: In my previous review, I suggested for estimation of Genotyping error (allele dropout and null allele). In the response, the author stated that they checked by Program micro checker and there was no sign of error. In this regard, the author needs to add a supplementary table for the output of Microchecker and add it to the revised manuscript.

Comments 2. Authors need to estimate pairwise relatedness using Queller and Goodnight relatedness estimators as implemented in GENALEX 6.41 and the maximum likelihood method as implemented in ML-RELATE. These estimates will give a better representation of relatedness levels in the population.

Reviewer #2: The authors had made all the necessary changes to the revised manuscript as suggested by the reviewers. The revised manuscript now looks more clear and can be published.

Even though, authors have explained well in detail regarding considering the loci with lower PIC values, I still believe the loci have limitation. Moreover, table 2 can be removed as the values are based on 55 loci which are not further used in any analysis.

7. PLOS authors have the option to publish the peer review history of their article (what does this mean?). If published, this will include your full peer review and any attached files.

Reviewer #1: No

Reviewer #2: **Yes: **Yellapu Srinivas

---

## [Editor Report · Acceptance letter]

25 Mar 2022

PONE-D-21-07853R1 

Development of 16 novel EST-SSR markers for species identification and cross-genus amplification in sambar, sika, and red deer 

Dear Dr. Ju:

I'm pleased to inform you that your manuscript has been deemed suitable for publication in PLOS ONE. Congratulations! Your manuscript is now with our production department. 

Kind regards, 

on behalf of

Dr. Mukesh Thakur 

Academic Editor

PLOS ONE